Mesenchymal stem cells alleviate sepsis-induced acute lung injury by blocking neutrophil extracellular traps formation and inhibiting ferroptosis in rats

Wang TieNan
Zhang Zheng
Deng Zhizhao
Zeng Weiqi
Gao Yingxin
Hei Ziqing heiziqing@sina.com
Yuan Dongdong yuandongdong123@126.com
Department of Anesthesiology, The Third Affiliated Hospital of Sun Yat-Sen University , GuangZhou, GuangDong Province , China
Uversky Vladimir
Electronic publication date: 2024 Jan 29
Publication date: 2024
Volume: 12
Electronic Location ID: e16748
Received 2023 Jun 22; Accepted 2023 Dec 12
Copyright: © 2024 Wang et al.
Copyright year: 2024
Copyright holder: Wang et al.
License: This is an open access article distributed under the terms of the Creative Commons Attribution License, which permits unrestricted use, distribution, reproduction and adaptation in any medium and for any purpose provided that it is properly attributed. For attribution, the original author(s), title, publication source (PeerJ) and either DOI or URL of the article must be cited.
License URL: https://creativecommons.org/licenses/by/4.0/

Keywords: Sepsis, Acute lung injury, Mesenchymal stem cells, Ferroptosis, Neutrophil extracellular traps

Funding: National Natural Science Foundation of China 82072216 & 81871597 Natural Science Foundation of Guangzhou 202102020167 Research Foundation of Guangdong Province 2019A1515010093, 2021A1515111153, 2022A1515012611 & 2022A1515011556 The National Natural Science Foundation of China U22A20276 Science and Technology Planning Project of Guangdong Province-Regional Innovation Capacity and Support System Construction 2023B110006 Science and Technology Program of Guangzhou, China 202201020429 The “Five and five” Project of the Third Affiliated Hospital of Sun Yat-Sen University 2023WW501 This work was supported by National Natural Science Foundation of China (No. 82072216, 81871597); Natural Science Foundation of Guangzhou City (No. 202102020167), Basic and Applied Basic Research Foundation of Guangdong Province (No. 2019A1515010093, 2021A1515111153, 2022A1515012611, 2022A1515011556). The National Natural Science Foundation of China (No. U22A20276), Science and Technology Planning Project of Guangdong Province-Regional Innovation Capacity and Support System Construction (No. 2023B110006), Science and Technology Program of Guangzhou, China (No. 202201020429) and the “Five and five” Project of the Third Affiliated Hospital of Sun Yat-Sen University (No. 2023WW501) paid for the APC and only the APC. The funders had no role in study design, data collection and analysis, decision to publish, or preparation of the manuscript.

==============================
Acute lung injury (ALI) is one of the most serious complications of sepsis, characterized by high morbidity and mortality rates. Ferroptosis has recently been reported to play an essential role in sepsis-induced ALI. Excessive neutrophil extracellular traps (NETs) formation induces exacerbated inflammation and is crucial to the development of ALI. In this study, we explored the effects of ferroptosis and NETs and observed the therapeutic function of mesenchymal stem cells (MSCs) on sepsis-induced ALI. First, we produced a cecal ligation and puncture (CLP) model of sepsis in rats. Ferrostain-1 and DNase-1 were used to inhibit ferroptosis and NETs formation separately, to confirm their effects on sepsis-induced ALI. Next, U0126 was applied to suppress the MEK/ERK signaling pathway, which is considered to be vital to NETs formation. Finally, the therapeutic effect of MSCs was observed on CLP models. The results demonstrated that both ferrostain-1 and DNase-1 application could improve sepsis-induced ALI. DNase-1 inhibited ferroptosis significantly in lung tissues, showing that ferroptosis could be regulated by NETs formation. With the inhibition of the MEK/ERK signaling pathway by U0126, NETs formation and ferroptosis in lung tissues were both reduced, and sepsis-induced ALI was improved. MSCs also had a similar protective effect against sepsis-induced ALI, not only inhibiting MEK/ERK signaling pathway-mediated NETs formation, but also alleviating ferroptosis in lung tissues. We concluded that MSCs could protect against sepsis-induced ALI by suppressing NETs formation and ferroptosis in lung tissues. In this study, we found that NETs formation and ferroptosis were both potential therapeutic targets for the treatment of sepsis-induced ALI, and provided new evidence supporting the clinical application of MSCs in sepsis-induced ALI treatment.

Introduction

Sepsis is a life-threatening multi-organ dysfunction caused by a response to infection (Reinhart et al., 2017). Secondarily, acute lung injury (ALI) is commonly observed in patients with sepsis, and is strongly associated with high morbidity and mortality (Prescott & Angus, 2018). The most severe form of ALI is acute respiratory distress syndrome (ARDS), which causes inflammation of the alveolar-capillary membrane and pulmonary edema. Current treatments for ALI include anti-inflammatory therapies, such as antioxidants, protease inhibitors, or antiproteases, and cell therapies, such as stem cells (Bosma & Lewis, 2007). Gene therapy and new therapies such as the isolation of cell microvesicles are also forms of ALI treatment (Hu et al., 2022; Zoulikha et al., 2022). However, because the mechanism of sepsis-induced ALI is still unclear, effective clinical treatment methods have not been developed. Therefore, it is necessary to clarify the mechanism of sepsis-induced ALI in order to explore potential therapeutic methods.

Polymorphonuclear neutrophils (PMNs) are the dominant inflammatory cells recruited to the lungs in sepsis, and are central to the pathogenesis of sepsis-induced ALI (Grommes & Soehnlein, 2011). PMNs assail microorganisms after migrating to the infectious site by degranulation, phagocytosis, or the release of NETs, which are a web of DNA material and antimicrobial proteins. At first, it was reported that the inhibition of NETs reduced ALI caused by intestinal ischemia reperfusion injury (Zhan et al., 2022). Recently, Alsabani et al. (2022) has shown that NETs formation leads to lung injury in human and murine sepsis, asserting that the reduction of NETs is a possible therapeutic approach for treating endotoxic shock-induced ALI. However, little evidence is available on how NETs cause ALI/ARDS in sepsis. A recent study confirmed that NETs-mediated ferroptosis of alveolar epithelial cells is important to sepsis-induced ALI (Zhang et al., 2022). Known as a kind of cell death caused by lipid peroxidation and disordered iron metabolism, ferroptosis has been reported to be essential in sepsis-induced ALI (Shimizu et al., 2022), making it a potential target for treatment.

Few pharmacologic therapies have demonstrated efficacy and low toxicity in treating sepsis-induced ALI. An exciting treatment strategy addressing this pressing need is the use of cell-based therapies, including MSCs. It has been reported that MSCs can potentially promote tissue repair in sepsis-induced ALI (Rojas et al., 2005). Although there is some understanding of the therapeutic mechanisms, many are still unclear and some controversy persists, such as the effects of MSCs on NETs, ferroptosis, ALI, or ARDS in sepsis. This results in limitations on the clinical applications of MSCs. Therefore, in this study, we focused on two major concepts: the possible mechanisms of sepsis-induced ALI mediated by NETs or ferroptosis, and the protective effects of MSCs in this process.

We hypothesized that MSCs alleviate sepsis-induced ALI by inhibiting NETs-mediated ferroptosis. The relationship between MSCs, NETs, and ferroptosis was studied.

Materials and Methods

Animals

Healthy SPF (Sprague-Dawley, SD) male rats (200–250 g) were purchased from Hunan Skarjingda Co., LTD. The rats were raised at a room temperature of 25 °C to 27 °C and provided with a basic diet for 1 week before the experiments. All animal care and experimental protocols in this study were approved by the Institutional Animal Care and Use Committee of the Laboratory Animal Center of South China Agricultural University (No. 2021D028). The rats were provided for laboratory animal use in accordance with National Institutes of Health Guidelines (NIH Publication 86-23 revised 1985 (no longer available, the currently available NIH Guidelines are NIH, 2011)).

Experimental design

A total of 60 rats were housed in filtered-air ventilated cages with free access to food and water in an environment with a 12-h light/dark cycle, at a temperature of 20–26 °C. These SD rats were obtained from Hunan SJA Laboratory Animal Co., Ltd. In the first experiment, the rats were randomized to the sham group (n = 6) or the CLP group (n = 6), and lung tissue samples were collected 24 h after surgery. Rats undergoing the sham operation were subjected to ventral midline incision, sterilized, and sutured, and given the same amount of fluid as the CLP group. In the second experiment, the rats were randomly assigned into three groups (n = 6 in each group), and the effects of ferrostatin-1, DNAse-1, and U0126 (a specific inhibitor of MEK1) were observed as follows: sham group, CLP group, and CLP + corresponding inhibitor treatment group. Ferrostatin-1 (1 mg/kg, Med Chem Express, HY-100579, Shanghai, P. R. China)/DNAse-1 (5 mg/kg, Beijing Solarbio Science & Technology Co., Ltd, 9003-98-9, Beijing, China)/U0126 (100 μg/kg, ABSIN, ABS810003, Shanghai, China). All inhibitors were configured according to the instructions, and the carrier referred to the corresponding solvent. The inhibitors mentioned above were administered by tail vein injection immediately after CLP modeling while the sham group and CLP group were treated with a solvent of inhibitors. In the third experiment, the rats were randomly assigned into three groups (n = 6 in each group). The effects of MSCs were observed as follows: sham group, CLP + vector group, and CLP + MSC group. MSC (5 * 106 in 1 ml saline, mubmx-01001, Cyagen, Guangzhou, China) was injected through the tail vein 1 h after CLP modeling. Rats were anesthetized as previously described in Yao et al. (2019). The remaining rats that survived 24 h after CLP modeling were euthanized using the same methods. Samples were then collected. Only rats that survived 24 h after modeling were included in the statistical analysis.

Rat cecal ligation and puncture model

All procedures involving animals in the rat caecal ligation and puncture model were performed in accordance with the guidelines for animal experiments and were approved by the Institutional Animal Care and Use Committee of South China Agricultural University (Approval No. 2022D010). In accordance with the results of the preliminary experiment, the cecal ligation and puncture of rats was modified to 75% ligation and two hole punctures.

Treatment of mesenchymal stem cells

Adult bone marrow mesenchymal stem cells purchased from Saiye were cultured in OriCell® adult bone marrow mesenchymal stem cell complete medium (Cyagen, HuxMA-90011, Guangzhou, China). The seventh passage was digested, collected, counted, and centrifuged. An appropriate amount of William’s solution was added, and the cells were resuspended to a concentration of 5 * 106 ml.

Inhibitor treatment

The ferroptosis inhibitor group was treated with ferrostatin-1 (1 mg/kg, Med Chem Express, HY-100579, Shanghai, China) immediately after CLP modeling. The NETs inhibitor group was treated with DNASE-1 (100 μg/kg, Beijing Solarbio Science & Technology Co., Ltd, 9003-98-9, Beijing, China) 1 h after modeling. The MEK/ERK pathway inhibition group was treated with U0126 (100 μg/kg, ABSIN, ABS810003, Shanghai, China) immediately after modeling.

Hematoxylin and eosin staining

As described in the experimental design, the rats were sacrificed 24 h after CLP modeling and their lung tissues were harvested. The left lung was used to determine the dry/wet ratio, and the right middle lobe was fixed with formaldehyde and embedded in paraffin for subsequent hematoxylin and eosin staining. The pathological changes in lung tissue were observed under a light microscope. Two pathologists scored sepsis-induced ALI based on the assessment criteria described in figure legend of Fig. 1.

Figure 1 Ferroptosis was involved in sepsis-induced ALI.

(A) HE staining of lung tissue from sham group and CLP group, scale bar = 50 μm. (B) Pathological score of lung injury. According to neutrophil infiltration, alveolar tissue fragments, pathological cell proliferation, and degree of congestion, each item was scored as three points (>50%), three points (25–50%), one point (0–25%), and zero points. (C) The dry to wet ratio of lung tissue was measured to compare lung-infiltrating humics (D) ELISA was performed for testing inflammatory factors. (E) Immunohistochemistry was performed to examine ACSL4 expression in rat lung tissue, scale bar = 50 μm. (F) The relative IOD value of ACSL4. Each slice was compared to the average value of the Sham group. (G) The protein expression level of ferroptosis indicators ACSL4, GPX4, FTH in lung tissue. (H) Semi-quantitative analysis of ACSL4, GPX4 and FTH expression in Western Blot data. (I) GSH, MDA and iron concentration. Each bar represents the mean ± SEM (n = 6 per group) *p < 0.05; **p < 0.01; ***p < 0.001 independent samples t-test with Tukey’s test.

Dry/wet ratio

The left lobe was used in the dry/wet ratio experiment. After dissection, the left hilus was ligated, and the entire left lobe was cut off and placed into a clean and weighed centrifuge tube to measure the wet weight. The left lobe was then placed in a Petri dish and put into a 60 °C oven to measure the dry weight after drying for 48 h. The ratio of wet weight to dry weight was recorded as the final result.

Immunohistochemistry

Paraffin-embedded blocks of the right middle lobe were sliced into 5 μm thick sections and stained with rabbit anti-rat ACSL4 (1:100, A6826, ABclonal, Woburn, MA, USA). The expression of ACSL4 in lung tissue was detected by HRP rabbit secondary antibody. The sections were deparaffinized with xylene, dehydrated with ethanol, and heated in 0.01 M citrate buffer (pH 6.0). Endogenous peroxidase activity was inactivated in 3% H2O2 for 15 min at room temperature. After the sections were incubated in blocking buffer (10% goat serum albumin), they were incubated with primary antibodies recognizing ACSL4 (1:100, A6826, ABclonal, Woburn, MA, USA) at 4 °C overnight. The tissues were then incubated with secondary anti-rabbit antibody-coated polymer peroxidase complexes (A0208, Beyotime, Nanjing, China) at room temperature. After the tissues were incubated with chromogenic substrate DAB (P0203, Beyotime, Nanjing, China) for 1 min, the slides were incubated with hematoxylin (C0107, Beyotime, Nanjing, China) for 10 s. The sections were washed in running water for 20 m. Five fields per slides were randomly chosen by the viewer and semi-quantifiedwith Image j software.

Immunofluorescence and immunofluorescence radiography

The right upper lobe was embedded with an Optimal cutting temperature compound (OCT) embedding agent and cut into 5-μm sections. Potential non-specific staining in the sections was blocked with 5% goat serum albumin and 0.1% Triton X-100 in PBS. Rabbit Histone H3Rb mAb (1:200, AB219407, Abcam, Cambridge, UK) and Mouse MPO Mb mAb (1:100, YM33964, Immunoway, Plano TX, USA) primary antibodies were used to bind antigens. Dylight488, Goat Anti Rabbit IgG/Dylight594 Goat Anti Mouse IgG secondary antibodies (1:500, RS23220; RS3608, Immunoway, Plano, TX, USA) were fluorescent-labeled to detect the expression of neutrophil extracellular traps. A laser confocal microscope (Zeiss, LSM880, Oberkochen, Germany) was utilized for observing the stained sections. The z-stack model was used to capture several pictures from different depths in the same area. A maximum efficiency overlay was used to establish a final picture. Image J1.48 (National Institutes of Health) was used to calculate the fluorescence intensity, area, and neutrophil count after Photoshop and standard image processing.

Immunoblotting

Western blotting was performed following standard procedures. Anti-ACSL4 antibody (1:1,000, A16848, ABclonal, Woburn, MA, USA), anti-ferritin Heavy Chain (FTH) antibody (1:1,000; 381204, Zen Bio), anti-GPX4 antibody (1:5,000; ab125066, Abcam, Cambridge, UK), anti-ERK1/2 antibody (1:2,000; YT1625, Immunoway, Plano, TX, USA), anti-Phospho-Erk1/2 antibody (1:2,000; O1923, Cell Signaling, Danvers, MA, USA), anti-MEK1/2 antibody (1:2,000; D1A5, Cell Signaling, Danvers, MA, USA), anti-Phospho-MEK1/2 antibody (1:2,000; 41G9, Cell Signaling, Danvers, MA, USA) and secondary antibody (1:2,000; Beyotime, Jiangsu, China) were used to detect protein expression. Anti-GAPDH (Zen Bio) was used at a 1:2,000 ratio. Images were acquired by a Tanon 5500 imaging system (Tanon, Shanghai, China). The images were scanned with the Image J scanning software, and the data was expressed as relative values to sham or control values.

Determination of iron ion, GSH, and MDA in tissues

An approximately 1mg tissue block was put into dry ice and mechanically homogenized using zirconia beads. Tissue iron concentration was detected according to the tissue iron ion concentration kit (Nanjing Jiancheng, China, A039-2-1) protocol. The concentration of GSH in the tissues was detected according to the GSH and GSSG test kit (Beyotime, China, S0053) protocol, and the concentration of MDA in the tissues was detected according to the MDA test kit (Beyotime, China, S0131S) protocol.

ELISA

An approximately 1mg tissue block was placed into dry ice and mechanically homogenized with metal beads in PBS medium. The concentrations of TNF-α, IL-6, IL-10, and myeloperoxidase (MPO) (Jingmei Biological, China, JM-01597R1) in the lung tissue homogenate were detected. We used MPO to represent NETs level in lung tissue.

Statistical analysis

Each biological experiment was replicated at least three times. Results were expressed as mean ± SEM. Differences between the two groups were analyzed by an independent samples t-test and one-way analysis of variance. A Tukey’s test was used for further comparison. The histopathological score did not fit a normal distribution and was analyzed by non-parametric tests. GraphPad Prism 6 software was used for statistical analysis of all experimental data, and p < 0.05 was used as the threshold for statistically significant differences.

Results

Ferroptosis involved in sepsis-induced ALI

Sepsis-induced ALI is characterized by pathological damage, pulmonary edema, and inflammation. It is manifested as higher histopathological scores, W/D ratio, tissue TNF-α levels, and IL-6 levels in the CLP group (Figs. 1A–1D). The protein expression of-achaete scute family BHLH transcription factor 4 (ACSL4), glutathione peroxidase 4 (GPX4), ferritin heavy chain (FTH), and the levels of glutathione (GSH), malondialdehyde (MDA), and Fe2+ in lung tissue are all important indicators that reflect the degree of ferroptosis (Dixon et al., 2012). We found that GSH and GPX4 expression decreased and FTH showed no statistically significant changes but slightly decreased in the CLP group, while all the other ferroptosis-related indicators mentioned above increased significantly in the CLP group. This indicates that ferroptosis became more severe in sepsis-induced ALI and might be a major factor in ALI development (Figs. 1E–1I). After the septic rats were treated with fer-1, a ferroptosis inhibitor, the expression of ACSL4, MDA, and cell-free Fe2+ in lung tissue decreased, while the expression of GPX4, FTH, and GSH increased (Figs. 2A–2D). Pathological damage, pulmonary edema, and inflammation all decreased (Figs. 2E–2H). These results suggest that inhibiting ferroptosis reduced sepsis-induced ALI.

Figure 2 Ferroptosis inhibitor Fer-1 alleviated ALI.

(A) ACSL4 Immunohistochemistry, scale bar = 50 μm. (B) The protein expression level of ACSL4, GPX4 and FTH level in lung tissue. (C) Semiquantitative analysis of ACSL4, GPX4 and FTH expression in Western Blot data. (D) GSH, MDA and iron concentration in rat lung tissue. (E) HE staining for Sham group, CLP group and group with Fer-1 treatment after CLP modeling, scale bar = 50 μm. (F) Pathological score of lung injury. (G) The dry to wet ratio of lung tissue. (H) The level of inflammatory factors in lung tissue. Each bar represents the mean ± SEM (n = 6 per group). ‘*’ Means the group is compared to Sham group; ‘#’ means the group is compared to CLP group, ‘ns’ means the group has no statistical differences compared to both Sham and CLP group. *p < 0.05 ; **p < 0.01; ***p < 0.001; #p < 0.05 ; ##p < 0.01; ###p < 0.001,one-way ANOVA with Tukey’s test.

Inducing NETs depletion suppressed ferroptosis

In recent years, several reports have stated that NETs are a new mechanism of alveolar epithelial cell injury (Scozzi et al., 2022). Therefore, we explored the effect of NETs on the lungs and their function in lung ferroptosis. Figures 3A–3C showed that a large number of NETs emerged in lung tissue in the CLP group. The representative indicators, such as MPO and Histone H3 were both elevated. After the treatment of DNase-1 to induce NETs depletion, ferroptosis in sepsis-induced ALI was suppressed, the expression of ACSL4 and the levels of MDA and cell-free Fe2+ in lung tissue decreased, and the downregulation of GPX4 and GSH expression caused by CLP was reversed (Figs. 3D–3H). However, FTH expression seemed to decrease after DNase-1 treatment (Fig. 3G). From these results, we concluded that NETs depletion could decrease lung ferroptosis.

Figure 3 NETs inhibitor suppressed NETs formation and inhibited ferroptosis.

(A) Confocal microscope photography, scale bar = 50 μm. (B) The number of positive MPO markers represented neutrophil infiltration. The formation of NETs was represented by the ratio of co-staining number to positive MPO number of represented NETosis. (C) ELISA represented semiquantitative NETs concentration in lung tissue. (D) ACSL4 Immunohistochemistry, scale bar = 50 μm. (E) The relative IOD value of ACSL4. Each slice is compared to the average value of Sham group. (F) The protein expression level of ACSL4, GPX4, FTH level in lung. (G) Semi-quantitative analysis of ACSL4, GPX4 and FTH expression in Western Blot data. (H) GSH, MDA and iron concentration in lung tissue. ‘*’ Means the group is compared to Sham group; ‘#’ means the group is compared to CLP group, ‘ns’ means the group has no statistical differences compared to both Sham and CLP group. *p < 0.05 ; **p < 0.01; ***p < 0.001; #p< 0.05 ; ##p < 0.01; ###p < 0.001, one-way ANOVA with Tukey’s test.

Suppressing the MEK/ERK pathway decreased NETs formation and ferroptosis-induced ALI in sepsis

Figure 4 shows the changes we detected in the MEK/ERK signaling pathway, which is considered to be the main cause of NETs formation (Hakkim et al., 2011). The results showed that the phosphorylation levels of MEK and ERK significantly increased in CLP rats, which could be reversed by U0126, an MEK inhibitor (Figs. 4A and 4B). More importantly, with the inhibition of the MEK/ERK signaling pathway by U0126, the formation of NETs was significantly suppressed and manifested as MPO and histone depression (Figs. 4D and 4E). Subsequently, we explored the effects of U0126 on ferroptosis and lung injury. The results demonstrated that after inhibition of the MEK/ERK signaling pathway by U0126 pretreatment, expression of ferroptosis-related indicators, such as ACSL4, MDA, and cell-free Fe2+ in lung tissue of CLP group, all decreased, but GPX4 and GSH expression increased (Figs. 5E–5H). There were no significant changes in FTH levels (Fig. 5G). Histopathological scores, the W/D ratio, tissue TNF-α, and IL-6 levels also decreased (Figs. 5A–5D).

Figure 4 MEK inhibitor U0126 inhibited MEK-ERK pathway activation and reduced NETosis.

(A) WB was used to detect the phosphorylation level of MEK and ERK. (B) IOD ratio of pMEK to MEK and pERK to ERK. (C) Immunofluorescence to examine NETosis, scale bar = 50 μm. (D) Neutrophil inflitration and the degree of NETosis. (E) Semiquantitative NETs concentration in lung tissue. Each bar represents the mean ± SEM (n = 6 per group). *p < 0.05 ; **p < 0.01; ***p < 0.001 one-way ANOVA with Tukey’s test.

Figure 5 MEK inhibitor U0126 attenuated ALI and inhibited ferroptosis.

(A) HE staining for Sham group, CLP group and group with U0126 treatment after CLP modeling, scale bar = 50 μm. (B) Pathological score of lung injury. (C) The dry to wet ratio of lung tissue. (D) The concentration of inflammatory factors in lungs. (E) ACSL4 Immunohistochemistry, scale bar = 50 μm. (F) The protein expression level of ACSL4, GPX4 and FTH in lung tissue. (G) Semi-quantitative analysis of ACSL4, GPX4 and FTH expression in Western blot data. (H) The concentration of GSH, MDA and iron in lung tissue. Each bar represents the mean ± SEM (n = 6 per group). ‘*’ Means the group is compared to Sham group; ‘#’ means the group is compared to CLP group, ‘ns’ means the group has no statistical differences compared to both Sham and CLP group. *p < 0.05 ; **p < 0.01; ***p < 0.001; ##p < 0.01; ###p < 0.001, one-way ANOVA with Tukey’s test.

MSCs improved lung tissue ferroptosis and alleviated sepsis-induced ALI via blocking and suppressing MEK/ERK pathway-induced NETs formation

The results above indicate that lung tissue ferroptosis mediated by MEK/ERK pathway-induced NETs may be an important factor in sepsis-induced ALI. Therefore, as shown in Figs. 6 and 7, we observed the role of MSCs on lung tissue ferroptosis and sepsis-induced ALI and determined whether their function was related to the above mechanism. The results in Fig. 6 showed that after injection of MSCs, the phosphorylation levels of MEK and ERK (p-MEK and p-ERK) significantly decreased in CLP rats (Figs. 6A and 6B) and their downstream NETs formation was also partially blocked (Figs. 6C–6E). Additionally, injection of MSCs caused the expression of some ferroptosis-related indicators, such as ACSL4, MDA, and cell-free Fe2+ in lung tissue of the CLP group, to decrease, but GPX4, FTH, and GSH expression increased (Figs. 7A–7D). With the depletion of NETs in lung tissue and the improvement of lung tissue ferroptosis, CLP-induced ALI was restored: histopathological scores, W/D ratio, tissue TNF-α, and IL-6 levels all decreased (Figs. 7E–7H).

Figure 6 MSC inhibited MEK/ERK pathway activation and suprressed NETosis.

(A) The phosphorylation level of MEK and ERK. (B) IOD ratio of pMEK to MEK, and pERK to ERK. (C) Immunofluorescence for examining NETosis, scale bar = 50 μm. (D) Neutrophil inflitration and the degree of NETosis. (E) Semiquantitative NETs concentration in lung tissue. Each bar represents the mean ± SEM (n = 6 per group). *p < 0.05 ; **p < 0.01; ***p < 0.001 one-way ANOVA with Tukey’s test.

Figure 7 MSC inhibited ferroptosis and alleviated ALI.

(A) ACSL4 Immunohistochemistry, scale bar = 50 μm. (B) The protein expression level of ACSL4, GPX4 and FTH level in lung tissue. (C) Semi-quantitative analysis of ACSL4, GPX4 and FTH expression in Western blot data. (D) The concentration of GSH, MDA and iron in lung tissue. (E) HE staining for Sham group, CLP group and group with MSC treatment after CLP modeling, scale bar = 50 μm. (F) Pathological score of lung injury. (G) The dry to wet ratio of lung tissue. (H) The concentration of inflammatory factors in lung tissue. ‘*’ Means the group is compared to Sham group; ‘#’ means the group is compared to CLP group, ‘ns’ means the group has no statistical differences compared to both Sham and CLP group. *p < 0.05 ; **p < 0.01; ***p < 0.001; ##p < 0.01; ###p < 0.001, one-way ANOVA with Tukey’s test.

Discussion

There are some major findings in this study. With the application of fer-1, Dnase-1, and U0126 to inhibit ferroptosis, NETs, and the MEK/ERK pathway respectively, we confirmed that the MEK/ERK pathway could induce NETs formation and subsequently result in lung tissue ferroptosis and ALI. We also provided a potential strategy to protect against sepsis-induced ALI, MSC injection, which not only inhibited the MEK/ERK pathway and its downstream NETs formation, but also reduced lung tissue ferroptosis and sepsis-induced ALI (Fig. 8).

Figure 8 Schematic representation of the effect of MSC treatment alleviated sepsis-induced ALI by inhibiting ferroptosis and netosis.

As a recently discovered form of regulated cell death triggered by erastin or RSL3, ferroptosis was first reported by Dixon et al. (2012). Ferroptosis has been proven to be critical in sepsis-induced organ injuries, including the heart, liver, and intestine. Several reports have concluded that the severity of these kinds of sepsis-induced organ injuries can be alleviated by the inhibition of ferroptosis (Li et al., 2021, 2020; Wang et al., 2020; Wei et al., 2020). Therefore, multiple researchers have considered ferroptosis inhibition as a potential strategy for organ protection. However, the role of ferroptosis in sepsis-induced ALI is contradictory. A study has reported that erastin treatment promoted ferroptosis, reducing the inflammatory response and sepsis development (Oh et al., 2019). Meanwhile, a different study has shown that ferroptosis promoted macrophages against intracellular bacteria (Ma et al., 2022). These contradictory results suggested that the function of ferroptosis in sepsis-induced ALI was still unclear. In this study, we found that a ferroptosis inhibitor treatment protected the lungs from sepsis-induced ALI, confirming that ferroptosis mainly has a negative effect on sepsis-induced ALI.

The elevated expression of FTH was conducive to the clearance of iron ions and important to ferroptosis protection (Blankenhaus et al., 2019), meaning FTH levels usually decrease in ferroptosis (Li et al., 2020). However, there were also reports stating that the expression of FTH1 is up-regulated in lipopolysaccharides (LPS)-induced sepsis modeling rats (Xiao et al., 2021). Further studies indicate that the increase of myeloid FTH aggravated sepsis-induced inflammation and organ injury (Zarjou et al., 2019). The Western Blot results reported in this study indicated there were no significant differences in the FTH levels between the CLP group and the sham group. The FTH expression of some individuals in the CLP group was even up-regulated. The FTH levels in the 6 h histone and cfDNA treatment group in our in vivo experiment were higher than in the Control group. However, when Histone treatment was prolonged, FTH levels decreased relative to the Control group (Supplemental 3D–3E). The differences in FTH variation in our study require further experimental research. As we know, PMNs play both vital and well-established roles in sepsis-induced ALI. Besides regulating ALI through the production of reactive oxygen species (ROS) or executing degranulation and phagocytosis, PMNs have been highlighted in ALI pathogenesis in another way, called NETs. NETs are composed of a mixture of neutrophil granule proteins, nuclear chromatin, and mitochondrial DNA that primarily absolve a defensive role against infection (Lood et al., 2016; McIlroy et al., 2014; Neubert et al., 2018; Urban et al., 2009; Yousefi et al., 2009). Excessive NETosis would promote microvascular dysfunction or even direct cellular/tissue injury (Silva et al., 2021). Consequently, high levels of NETs in bronchoalveolar or peripheral blood are generally associated with the worst ARDS outcomes (Mikacenic et al., 2018; Pan et al., 2017). Although both NETs and ferroptosis have been proven to play important parts in sepsis-induced ALI, the relationship between them is still ambiguous. As we mentioned above, NETs have recently been reported to regulate sepsis-associated ALI by activating ferroptosis in alveolar epithelial cells (Zhang et al., 2022). Our study confirmed that NETs developed excessively through activation of the MEK/ERK pathway in the pathogenesis of sepsis-induced ALI, and the impairment of NETs was partly relative to ferroptosis.

MSCs can be isolated from many types of mesenchymal tissues, such as bone marrow, umbilical cord blood, adipose tissue, and placenta (Chamberlain et al., 2007). MSCs have been reported to mediate potent immunomodulation to influence both innate and adaptive immune cells. This represented an important mechanism underlying the benefits of cell-based treatment for sepsis-induced ALI, rather than the paradigm of trans-differentiation and cell replacement. Although the possible benefits of using MSCs to treat sepsis-induced ALI have already been investigated, the concrete mechanisms remained vague (Lee et al., 2011), severely limiting its clinical application for treating sepsis-induced ALI.

Our study also has limitations. Only in vivo experiments were carried out in this study, and the molecular mechanism was less involved. Some supplementary cell experiments have been performed, but they were not sufficient. Additionally, one inhibitor was used in some experiments, and the specificity was relatively insufficient. More effort should be put into demonstrating how MSC therapy affects NETs formation.

Several previous studies, including ours, have clarified that both NETs and ferroptosis are critical to the pathogenesis of sepsis-induced ALI (Qu et al., 2022; Qu et al., 2021; Scozzi et al., 2022; Zhang et al., 2021). However, the relationship between MSCs, NETs, and ferroptosis remained unclear. In this study, we showed that MSC injection significantly alleviated sepsis-induced ALI. We also demonstrated that the underlying mechanism of sepsis-induced ALI may be related to MSCs blocking the formation of NETs and their downstream ferroptosis through inactivating the MEK/ERK pathway. We believe that our research not only clarifies the mechanism of sepsis-induced ALI but also provides a new theoretical basis for the clinical application of MSCs.

Conclusions

Our findings demonstrated that MSCs could be used as a therapy to effectively alleviate sepsis-induced ALI. The underlying mechanisms of this potential treatment method block the formation of NETs and inhibit ferroptosis.

Supplemental Information

Supplemental Information 1 ARRIVE 2.0 Checklist.

Click here for additional data file.

Supplemental Information 2 Raw data for Figure 1.

Click here for additional data file.

Supplemental Information 3 Raw data for Figure 2.

Click here for additional data file.

Supplemental Information 4 Raw data for Figure 3.

Click here for additional data file.

Supplemental Information 5 Raw data for Figure 4.

Click here for additional data file.

Supplemental Information 6 Raw data for Figure 5.

Click here for additional data file.

Supplemental Information 7 Raw data for Figure 6.

Click here for additional data file.

Supplemental Information 8 Raw data for Figure 7.

Click here for additional data file.

Supplemental Information 9 Figure 1 WB blot.

Click here for additional data file.

Supplemental Information 10 Figure 2 WB blot.

Click here for additional data file.

Supplemental Information 11 Figure 3 WB blot.

Click here for additional data file.

Supplemental Information 12 Figure 4 WB blot.

Click here for additional data file.

Supplemental Information 13 Figure 5 WB blot.

Click here for additional data file.

Supplemental Information 14 Figure 6 WB blot.

Click here for additional data file.

Supplemental Information 15 Figure 7 WB blot.

Click here for additional data file.

Supplemental Information 16 Extra data of figures.

(A) Grayscale statistic of immunohistochemistry picture in figure 1. (*compared with Sham group, ***, p＜0.001) (B) Grayscale statistic of immunohistochemistry picture in figure 2. (*, p＜0.05; ns, no statistical significance) (C) Grayscale statistic of immunohistochemistry picture in figure 3. (***, p＜0.001) (D) Grayscale statistic of immunohistochemistry picture in figure 5 (*, p＜0.05; **, p < 0.01) (E) Grayscale statistic of immunohistochemistry picture in figure 7 (*, p＜0.05) (F) Nets formation in Sham, CLP, Dnase-1, Fer-1 group.

Click here for additional data file.

Supplemental Information 17 Acquisition of cfDNA from NETs induced by human blood neutrophils in vitro.

(A) Reichsen-Giemsa staining of neutrophils extracted. Arrows indicate cells with distinct lobed nuclei. (B) Field of view of DAPI stained neutrophils in Control group and PMA 6h-induction group. (4X objective magnification) (C) Immunofluorescence of NETs formation. (10X objective magnification) (D) cfDNA concentration of culture medium in control group(DMSO group), PMA 6h-induction group, PMA 24 h-induction group, LPS 6h-induction group and LPS 24h-induction group. (***, p < 0.001) (E) cfDNA concentration of washing culture medium in control group(DMSO group), PMA 6h-induction group, PMA 24h-induction group, LPS 6h-induction group and LPS 24h-induction group. (*, p < 0.05; ***, p < 0.001)

Click here for additional data file.

Supplemental Information 18 Effects of NETs components on lung epithelial cells.

(A) Representative images of BESA-2E cells. (4X objective magnification) (B) Representative images of BESA-2E cells. (10X objective magnification) (C) Cell viability of BESA-2E cells after treatment of NETs components. (D) Western Blot of Ferroptosis marker protein expression of BESA-2E cells after treatment of NETs components for 6 h. Histone1 group was treated with 10 ug/ml histone for 6 h, Hisone2 group was treated with 10 ug/ml histone for 24 h. (E) Semi-quantitative figure of Western Blot.

Click here for additional data file.

Supplemental Information 19 Raw data for Figure S1.

Click here for additional data file.

Supplemental Information 20 Raw data for Figure S2.

Click here for additional data file.

Supplemental Information 21 Raw data for Figure S3.

Click here for additional data file.

Supplemental Information 22 Death rate of animal modeling.

Click here for additional data file.

Supplemental Information 23 Western blot picture for Figure S3.

Click here for additional data file.

Supplemental Information 24 Supplemental Methods.

Click here for additional data file.

Additional Information and Declarations

Competing Interests

Author Contributions

Animal Ethics

Data Availability

The authors declare that they have no competing interests.

TieNan Wang performed the experiments, prepared figures and/or tables, and approved the final draft.

Zheng Zhang analyzed the data, authored or reviewed drafts of the article, and approved the final draft.

Zhizhao Deng performed the experiments, prepared figures and/or tables, and approved the final draft.

Weiqi Zeng analyzed the data, prepared figures and/or tables, and approved the final draft.

Yingxin Gao analyzed the data, prepared figures and/or tables, contributed reagent, and approved the final draft.

Ziqing Hei conceived and designed the experiments, authored or reviewed drafts of the article, and approved the final draft.

Dongdong Yuan conceived and designed the experiments, authored or reviewed drafts of the article, and approved the final draft.

The following information was supplied relating to ethical approvals (i.e., approving body and any reference numbers):

Institutional Animal Care and Use Committee of the Laboratory Animal Center of South China Agricultural University (Guangzhou, China).

The following information was supplied regarding data availability:

The raw data is in the Supplemental Files.

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
