# Peer review of "Mesenchymal stem cells alleviate sepsis-induced acute lung injury by blocking neutrophil extracellular traps formation and inhibiting ferroptosis in rats"

_PeerJ, doi:10.7717/peerj.16748_

## Round 0.1 · original submission · Major Revisions

Please address the concerns of all reviewers and revise the manuscript accordingly.

**Language Note:** PeerJ staff have identified that the English language needs to be improved. When you prepare your next revision, please either (i) have a colleague who is proficient in English and familiar with the subject matter review your manuscript, or (ii) contact a professional editing service to review your manuscript. PeerJ can provide language editing services - you can contact us at copyediting@peerj.com for pricing (be sure to provide your manuscript number and title). – PeerJ Staff

Reviewer 1 ·

Basic reporting

This manuscript explores the role of Mesenchymal Stem Cells (MSCs) in a sepsis-induced acute lung injury model, focusing on their effects on Neutrophil Extracellular Traps (NETs) formation and ferroptosis. The detailed data presented in this research could lead to intriguing insights, particularly if MSCs could be potentially utilized in the treatment of Acute Lung Injury (ALI).

Experimental design

However, the manuscript primarily discusses the involvement of ferroptosis and NET formation in ALI, which is already extensively demonstrated in other works (Jhelum et al., 2018; Huang et al., 2021; Scozzi et al., 2022; Lai et al., 2023).

Validity of the findings

The authors' main effort to illustrate how and why their cell therapy is effective appears insufficient. Additionally, the paper lacks concrete experiments demonstrating how NETs formation in neutrophils could influence ferroptosis in alveolar epithelial cells.

Annotated reviews are not available for download in order to protect the identity of reviewers who chose to remain anonymous.

Reviewer 2 ·

Basic reporting

In the present manuscript, authors have studied the therapeutic potential of mesenchymal stem cells (MSCs) in improving sepsis induced acute lung injury. Using in vivo studies, their findings reveal that this protective effect of MSCs is mediated via inhibition of MEK/ERK signaling pathway which further inhibits neutrophil extracellular traps (NETs) formation and alleviation of ferroptosis.
The manuscript is well written but there are a few grammatical errors throughout the manuscript. The abstract and background clearly states the rationale for the present study. The methods and results are presented in sufficient details. However, the discussion section needs to be strengthened further more by providing additional information on following points:

• In the discussion section line 292: when you state that the role of ferroptosis is contradictory in sepsis induced-ALI, kindly elaborate on what are those contradictions, since using some very straightforward experiments, negative association of ferroptosis is shown with ALI.
• On line 327, you have provided a few references that have already proven involvement of NETs and ferroptosis in ALI, so in terms of role of NETs and ferroptosis in ALI, the study does not provide any additional piece of information.
• For delineating the protective role of MSCs in sepsis induced ALI, a thorough investigation should be conducted to understand the detailed molecular mechanism of action since the role of MSCs in improving survival in ALI through inhibition of NETs formation (Pedrazza et al., 2017) is already reported, also protective role of MSCs against abdominal aortic aneurysm formation by inhibiting NET-induced ferroptosis are known (Chen et al., 2023).
• Discussion section should elaborate on the advantage of using MSCs as therapeutic agent for ALI as opposed to other treatment strategies.
• The limitations and future directions of the study need to be discussed.

Experimental design

-

Validity of the findings

-

Additional comments

-

Reviewer 3 ·

Basic reporting

The paper is well written with sufficient literature background and article structure.
The results are relevant with the experimental design.

Experimental design

The research paper is well written with proper experimental design and the details are presented well.

Validity of the findings

Rationale is clearly stated, and the analysis are well defined..

Annotated reviews are not available for download in order to protect the identity of reviewers who chose to remain anonymous.

---

## Round 0.2 · accepted · Accept

All concerns are adequately addressed, and amended manuscript is acceptable now.

Reviewer 1 ·

Basic reporting

The authors have carefully addressed all my comments.

Experimental design

The manuscript is ready for publication.

Validity of the findings

See above.

Reviewer 3 ·

Basic reporting

The proficiency of entire manuscript is good. The authors provided enough literature review. All the figures and tables are alligned.

Experimental design

The methods are in details and well structured.

Validity of the findings

THe findings were presented appropriately and the conclusion is good.